# Wood Fuel Procurement to Bioenergy Facilities: Analysis of Moisture Content Variability and Optimal Sampling Strategy

**Elena Leoni [1], Manuela Mancini [2], Giovanni Aminti [3] and Gianni Picchi [3,\*]**

1 Department of Agricultural, Food and Environmental Sciences, Università Politecnica delle Marche, via Brecce Bianche, 60131 Ancona, Italy; e.leoni@pm.univpm.it

2 Department of Food Science, University of Copenhagen, Rolighedsvej 26, DK-1958 Frederiksberg, Denmark; manuela@food.ku.dk

3 CNR-IBE (Institute of Bioeconomy of the National Research Council), via Madonna del Piano 10, 50019 Sesto Fiorentino, Italy; giovanni.aminti@ibe.cnr.it

* Correspondence: gianni.picchi@cnr.it

**Abstract:** Moisture content is the most relevant quality parameter for wood fuels. Effective and fast determination of moisture of incoming feedstock is essential in the management of bioenergy facilities. The availability of fast and reliable moisture meters based on innovative technologies simplifies this task. However, in Mediterranean conditions the inherent variability of wood fuels calls for a careful sampling strategy if representative results are required while facing acceptable analytic costs. The present study is aimed at measuring the fuel heterogeneity and defining accordingly the appropriate number of samples to be analyzed in order to get reliable moisture-content results. A total of 70 truckloads (about 2270 t of woodchips) were sampled during commercial operations in two different seasons. Five samples were collected from each load and measured with standard method and magnetic resonance gauge. Results show that the variability of moisture content is influenced by mixing of species and storage of biomass. Heterogeneity can vary greatly also within single truckloads, to the point that three samples are needed to achieve about 90% of estimates within the desired precision limits. In the case of larger lots, such as barge or ship loads, 20 samples can provide sufficient precision in most scenarios.

**Keywords:** sampling; fuel; moisture content; quality; heterogeneity; biomass

## 1. Introduction

The European Union is promoting the substitution of fossil fuels with renewable energy sources with an ambitious plan: by 2030 it aims to reduce at least 40% of greenhouse gas emissions and to cover 27% of energy consumption with renewable energy sources [1]. Even more ambitious is the target to reduce by 2050 the greenhouse gas emissions by 90% as compared to 1990 [2]. These policies have boosted the development of renewable energies. Among those, bioenergy based on woody fuels is still considered one of the main contributors thanks to the availability of a large provision of unused feedstock resources, such as forest logging residues [3] and agricultural residues, industrial and urban waste and, in some contexts, dedicated energy crops [4]. In Italy the number of biomass-fed power plants increased from 75 to 89 in 2012–2013 [5] and is still a thriving sector. Yet, the economic viability and environmental impact of bioenergy production in this country are highly influenced by the great heterogeneity of qualitative properties and energetic characteristics of local wood fuels [6,7].

In this frame, an effective quality control becomes a key factor for the supply chain [8]. Among all quality parameters, Moisture Content (MC) is considered the most relevant [9,10] as it may affect the whole supply chain up to energy conversion. The MC of woody fuels depends on several aspects such as species, inclusion of different tree parts (presence

of leaves/needles) and season, among others [11,12]. Furthermore, MC can be influenced by the harvesting and storage operations [13]. The detrimental effects of MC can be highlighted along the whole supply chain [14]. Transport efficiency is penalized by the higher share of moisture over the total payload in truck transportation [15], due to its influence on the bulk density [16]. During yard storage in bioenergy facilities, a high MC promotes microbial activity, which in combination with long storage, large wood-chip piles and adverse weather conditions, may lead to significant dry-matter losses [9]. These can be partially reduced with sheltering systems for the piles, yet this solution increases storage costs and may easily offset the benefits obtained [17]. Moreover, in certain conditions the presence of soluble nutrients and high nitrogen content boosts the microbiological activity [13], which may quickly increase the temperature of biomass. In the worst cases it leads to self-ignition, causing immediate loss of fuel or, at the least, increases the ash content, reducing the overall energy content of the feedstock [18]. As a final drawback, MC is negatively correlated to the heating value, reducing the energy content and the market value of the feedstock [5]. Net calorific value could vary from 6 MJ/kg to 12 MJ/kg in woodchip with a range of moisture content from 2 to 58% [19], while with a mean value of moisture content about 10%, mean gross calorific value is about 18 MJ/kg [20]. In fact, with a high share of water over the dry biomass, the evaporation process subtracts energy from the combustion, reducing the temperature of the process [21]. This causes a non-decrease of the heating value, which drops from 14.07 to 7.82 Mj/kg, respectively, with a moisture content of 20 and 50%. Finally, the lower combustion efficiency caused by a high MC increases the pollutant emissions released with the flue gases, causing a significantly higher environmental impact [22].

Due to the importance of MC in biofuels, it is common practice of bioenergy facilities to set price classes for the incoming biofuel with monetary value inversely proportional to MC. This solution promotes the provision of higher-quality fuel but requires the technical capacity to assess the MC of the incoming loads. Ideally, this measurement should be performed quickly and with a reliable system, allowing the immediate refusal of loads exceeding the agreed threshold. This is hindered by the time required to determine the MC of woody biomass according to the standard method ISO 18134-1:2015 (up to 48 hours in oven).

As a response, in the past two decades several producers introduced in the market a number of moisture meters based on diverse technologies and types of sensors. After a setting-up period, the newest MC gauges proved comparable to the standard method in terms of precision and accuracy, but with a much faster production of results and thus capable of handling a higher number of samples in a given time. Nystrom and Dahlquist [23] compared several alternative technologies with the standard oven dry method, concluding that Near Infrared (NIR) spectroscopy and Radio Frequency (RF, also known as dielectric) technologies were the best-suited for MC measurement in flow and bulk fuels, respectively. More recently, further studies confirmed the potential of NIR instruments for MC determination due to their real-time and intuitive approach and the possibility to measure samples without any specific preparation or alteration of the biomass [24], allowing reiterate analysis on the same sample, if required. However, fuel-specific calibration is necessary for correct NIR deployment, making it a less flexible option in conditions of high variability of the biomass characteristics [19]. A further technological option recently introduced among commercial MC analyzers is Magnetic Resonance (MR) [25]. This was successfully tested in industrial environment for quick determination of MC of incoming trucks [22], proving capable to perform about 130 analyses in an 8-hour shift and with no sensitivity to the heterogeneity of woody biomass and no need of fuel-specific calibration models.

The latter aspect may be crucial for future developments in the field of industrial MC determination of biofuels. In fact, with high heterogeneity of wood species and/or intrinsic characteristics of the incoming biomass, an improper sampling technique could lead to significant errors even with modern sensors, particularly in the case of large consumers

of fuel [18]. In fact, the selection of a limited amount of material to be regarded as physically and chemically representative of large quantities of biomass is a challenging and uncertain operation [26]: in case of inappropriate sampling the whole MC determination process could be biased. This problem is particularly relevant in areas with high heterogeneity of woody fuels, as may be the case with systems based on Mediterranean mixed forests, whose degree of variability is largely unknown.

The technical standard ISO 18135 [27] reports a sampling procedure based on the type and variability of the biofuel, to be assessed beforehand or according to expert assessment, and provides tables to calculate the appropriate sampling intensity. Nevertheless, the application of the guidelines of the standard in commercial biomass sampling showed some critical issues, particularly regarding the excessive demand in terms of time and effort required. Furthermore, the heterogeneity of woody biofuel could be overlooked by the limited availability of consultative tables, which are related just to major biomass typologies. Therefore, guidelines often return a sampling strategy excessively demanding and not linked to the actual uncertainty degree of the MC estimate at the power plant. [26].

The possibility to perform reliable analysis in a short time and within the premises of the power plant paves the way to a new approach of MC monitoring of incoming biomass with an increased number of samples per load received and a better description of the fuel characteristics. But this approach requires a deeper understanding of the intrinsic quality variability of woody fuels and the most appropriate sampling procedures to be adopted accordingly.

Against this background the present study aimed at:

1     better understanding the range of MC variability of incoming biomass in a power plant in real conditions;

2     relating the MC variability with the reliability of moisture estimate performed on site with a Magnetic Resonance moisture meter (MR-MM), and;

3     assessing the optimal number of samples to be collected for MC determination (sampling intensity) in the cases of incoming high-volume trucks and of larger lots of woody fuel.

## 2. Materials and Methods

Samples were collected at the fuel yard of a biomass power plant located in a mountain area of Southern Italy in two different sampling periods, hereafter identified as "Summer" and "Autumn". These were selected in order to catch the maximum possible variability in a short time span. Summer sampling was done in early October, when contractors delivered both biomass from ongoing harvest operations (thus with high MC) and material left to dry in intermediate yards during the warm and dry summer. Autumn sampling, performed in late November, involved just wet biomass since dry stocks were depleted and the Mediterranean rainy season had commenced.

The power plant of this study was selected also for the local high variability of tree species and forest treatments that provide the feedstock to the facility. The main forest species locally delivered as wood chips are chestnut, deciduous oaks, poplar, willow and alder among broadleaves; fir, pine and occasionally cypress among conifers. Due to the dominance of mixed forests and the variability of work systems deployed, the biomass can be either mono-specific or a mix of several species. Additionally, procurement can be based on recently felled and comminuted trees, or biomass stored in satellite yards either as logs or loose wood chips. All these factors contribute to the high variability of incoming feedstock and were duly noted as reference data for each truckload in order to analyze their influence on the overall heterogeneity and the reliability of MC fast estimation.

Wood chips were delivered to the power plant with high-volume, moving-floor semitrailers carrying an average payload of 32.4 tonnes (t). A total of 35 trucks were sampled per season, for a total of 70 incoming trucks involved in the study delivering about 2270 t of wood chips. Biomass collection was performed immediately after unloading the wood

chips on the paved yard. Five replicates were sampled per pile in predefined positions chosen in order to describe the possible internal variability: three collecting spots were identified on the top and two at the bottom of the heap of wood chips (Figure 1). A manual grain scoop was used to collect the biomass, filling 5 buckets with a capacity of 10 dm³ each.

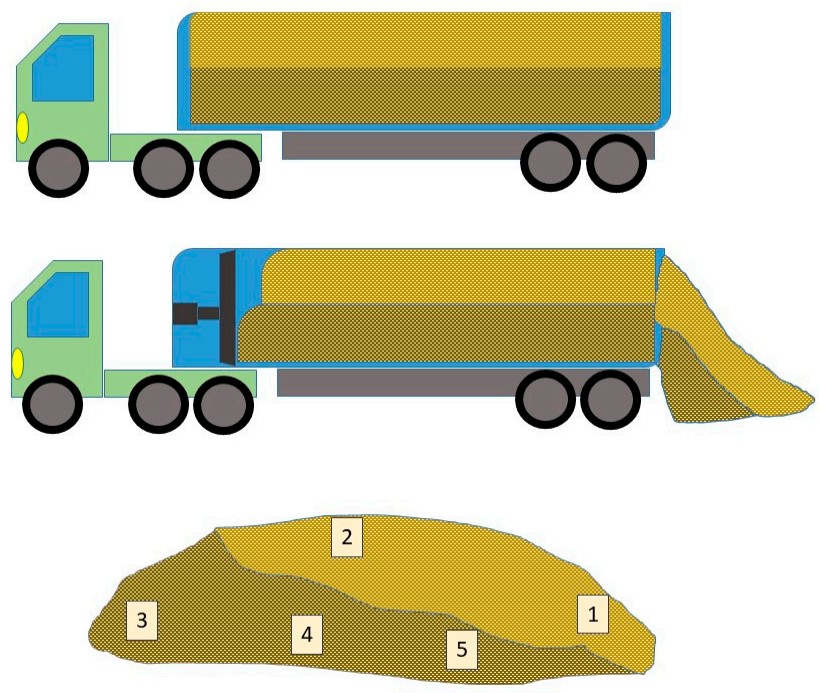

**Figure 1.** Location of the predefined sampling areas in the pile of unloaded wood chips (side view).

At the control station of the biomass yard, the content of each bucket was carefully mixed prior to the extraction of 2 subsamples from each sample (Figure 2). The first subsample (hereafter named STAN) had an approximate volume of 3 dm³. It was tipped in a paper bag of known tare and weighed with a precision scale for wet mass determination. Afterwards, all bags were closed and transported to a laboratory to determine the oven dry mass according to standard method (ISO 18134-1:2015). A laboratory scale with a precision of 0.01 g was used for weighing all subsamples.

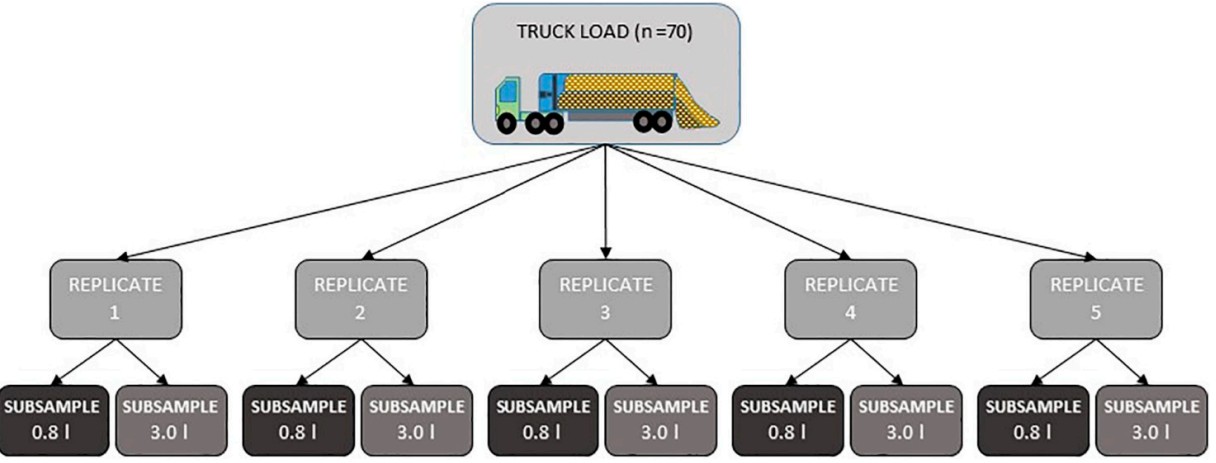

**Figure 2.** Sampling strategy for the selection of 5 first subsamples (STAN) and 5 second subsamples (FAST) from each incoming truck.

The second subsample (hereafter named FAST) was collected from the bin using a 0.8 dm³ plastic container, which was directly introduced in the MR–MM for MC determination. Overall, 700 subsamples equally divided between STAN and FAST were collected and analyzed. Further details regarding sampling procedure and the characteristics and performance of the MR–MM can be found in [19].

### 2.1. Sampling Intensity on Trucks

This action aimed at assessing the MC estimate improvement related to the analysis of an increasing number of samples from the same load (sampling intensity). The average MC of all the five STAN samples collected on each truckload was considered as the reference value (REF_MC). This was compared with the average MC of the FAST samples (MR_MC). The different levels of sampling intensity of MR_MC was simulated using an increasing number of FAST values randomly chosen within the same truckload. For this purpose, a research randomizer [28] was deployed to casually select the MC values to be used for each comparison. The simulated sampling intensity ranged from 1 to 4 FAST values used to calculate the average MC: the combination of the 5 FAST values was not considered since no randomization was possible. In order to avoid random effects, each sampling intensity level (1 to 4) was replicated 5 times with a new set of randomized values. The average values of the 5 replications were used to calculate the error according to the following formula:

$$\delta_{X,I} = REF\_MC_x - MR\_MC_{x,I}$$

(1)

Where:

"X" is the truckload, ranging from 1 to 70

"I" is the number of random samples used to calculate the average MC, ranging from 1 to 5

"REF_MC" is the MC of the truckload X

"MR_MC" is the average MC of the "I" samples randomly selected among the 5 measured on the truckload X

### 2.2. Sampling Intensity on Piles

Similarly to the above procedure, sampling intensity was simulated for large piles of wood chips. The analysis was performed assuming to have two piles composed by the 35 truckloads of each season, thus with an approximate mass of 1130 t. The MC of the piles was calculated with the average value of the 175 STAN samples collected and measured in each season. Estimation of the MC was simulated as the average of a variable number of FAST values. The number of samples used to calculate the average varied according to the assumed sampling intensity: 5, 10, 20 and 30 samples per pile. The simulation was accomplished again with randomization of the FAST values, generating 10 sets of randomly arranged MC values for the two datasets (Summer and Autumn). From these datasets, 50 subsets (replications) of 5, 10, 20 and 30 MC values were randomly selected, and the corresponding average value was calculated. The MC estimation error for the two piles/seasons was calculated according to the following formula:

$$\Delta_k = REF\_MC - MR\_MC_k$$

(2)

Where:

"k" is the number of random samples used to calculate the average MC, with values set to 5, 10, 20 and 30

"REF_MC" is the MC of the biomass pile

"MR_MC" is the average MC of the "k" samples randomly selected among the 175 measured on the truckloads composing the pile

## 2.3. Statistical Analysis

Multifactor ANOVA analysis of MC values was performed to verify normal distribution of data and influence of season and external factors on MC. The five MC values collected in each truckload were used to calculate an average MC value for each sample and the standard deviation of each load ($SD_t$). ANOVA and descriptive analysis were used to compare the values of MC and $SD_t$ of trucks and the effect of season and other factors on the variability. An F-test was performed to compare SD of the average MC and $SD_t$ values of trucks in different seasons.

## 3. Results

### 3.1. Fuel Quality and Heterogeneity

Moisture analysis of STAN samples confirmed the difference expected between the two sampling seasons. In spite of the limited timespan, the average moisture content rose about 4% in the loads sampled in the Autumn sampling season, reaching almost the 50% threshold (Table 1). It is worth mentioning that companies acted in the frame of a contract setting 45% MC as the target value for the delivered biomass; thus, it is reasonable to assume that the operators made all possible efforts to counter this variation, to be regarded as a consequence of the Mediterranean rainy season. The SD of the whole dataset and the SD of the average MC values of single truckloads (REF_MC) are higher for the Summer period compared to Autumn, even if there is no statistical difference according to the variance check with Levene's test. Accordingly, the range of MC values is much higher in Summer, with a difference of more than 15 percentage points both when considering the whole dataset and the average values of truckloads.

**Table 1.** Moisture content (MC) of all samples for the two seasons. Values in column with different letters (a and b) show a statistical difference with P-Value below 0.05.

| Sampling Season | Avg. Moisture Content (MC%) | Standard Deviation | Coefficient of Variation (%) | Minimum Value (MC%) | Maximum Value (MC%) | Range (MC% Points) |
|---|---|---|---|---|---|---|
| ALL SAMPLES | | | | | | |
| Summer | 44.5 [a] | 6.01 | 13.51 | 24.9 | 60.9 | 36.0 |
| Autumn | 49.2 [b] | 4.88 | 9.91 | 39.4 | 58.8 | 19.4 |
| AVERAGE VALUE OF TRUCKS | | | | | | |
| Summer | 44.5 [a] | 5.84 | 13.1 | 27.2 | 56.3 | 29.1 |
| Autumn | 49.2 [b] | 4.60 | 9.35 | 42.4 | 57.5 | 15.1 |

By sorting the truckloads according to the increasing REF_MC value (Figures 3 and 4), it is possible to better visualize the variability recorded among and within the loads. Neither the two sampling seasons nor the REF_MC values show a clear influence on the heterogeneity of the feedstock composing the load. This was confirmed by the F-test comparing SD values of the single trucks (Table 2), where no significant difference was detected among the averages of SD values. This confirms that the internal variability of the biomass carried in each truck did not change with the season and remains very high with a 4.5 range of SD values.

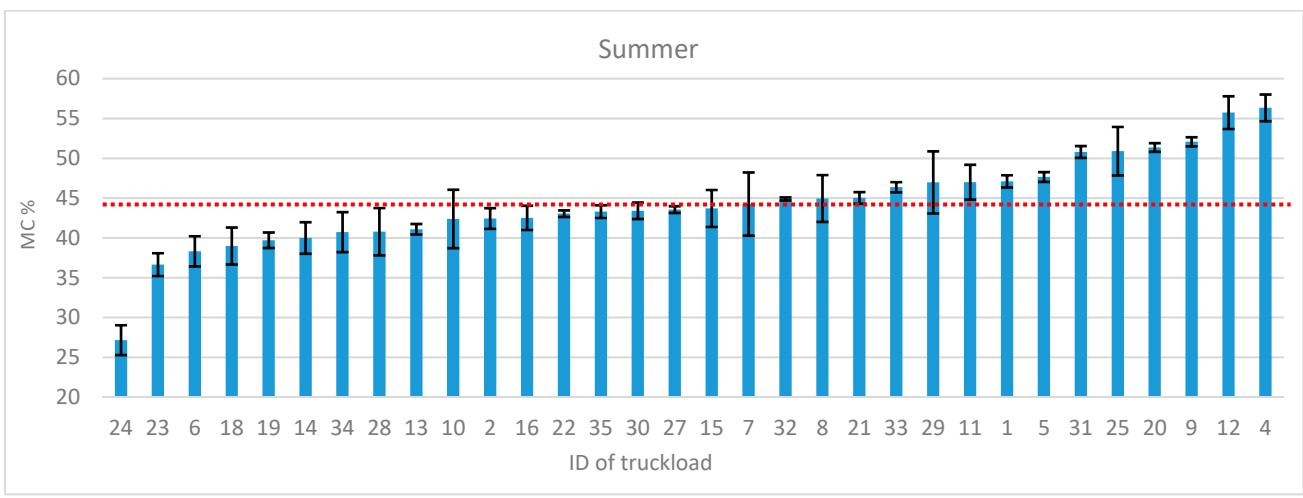

**Figure 3.** Average MC values of truckloads in increasing order for Summer sampling period. The red line represents the average MC value (44.3%), black bars show the SD.

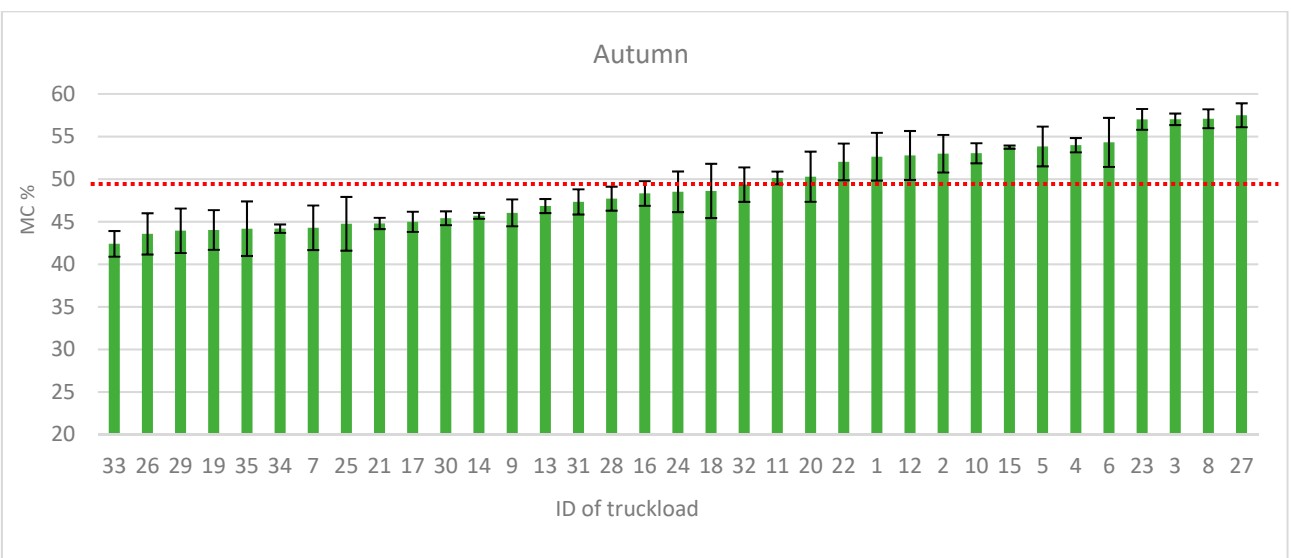

**Figure 4.** Average MC values of truckloads in increasing order for Autumn sampling period. The red line represents the average MC value (49.2%), black bars show the SD.

**Table 2.** Analysis of the SD values of each trucks (relative to the 5 samples measured in each truck).

|  | *Summer* | *Autumn* |
|---|---|---|
| Count | 35 | 35 |
| Average | 1.94 | 2.08 |
| Standard deviation | 1.21 | 1.09 |
| Coeff. of variation | 62.2% | 52.7% |
| Minimum | 0.51 | 0.48 |
| Maximum | 5.03 | 4.88 |
| Range | 4.52 | 4.40 |

In order to better describe the heterogeneity of the incoming truckloads, $SD_t$ values were ranked in unitary SD classes, providing their frequency distribution (Figure 5). Assuming that the desired degree of MC variability for an industrial biomass user falls below an SD value of 2, the threshold is exceeded by 31.4% and 40% of the loads, respectively, for the Summer and Autumn periods.

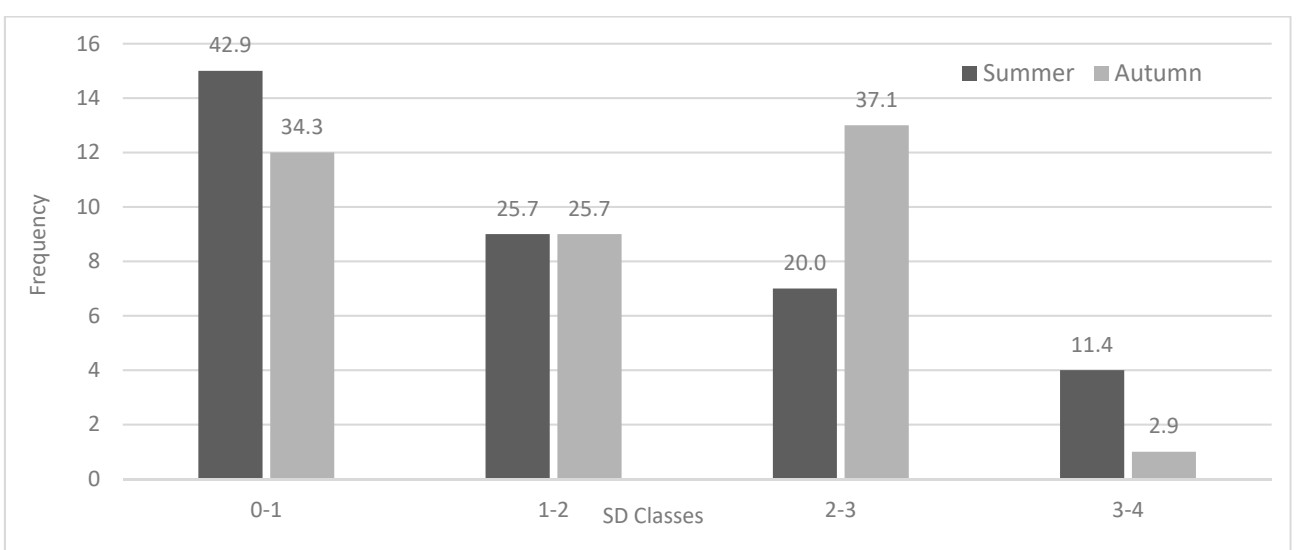

**Figure 5.** Classes of SD values of the loads measured in the two sampling seasons.

The analysis of external factors showed that the intermediate storage of biomass prior to transportation has a high impact on $SD_t$ values, with a significant increase when biomass is stored in satellite yards before its delivery to the power plant (Figure 6). The species composition did not result in a statistically significant effect on $SD_t$, yet the means plot table shows that pure material either from conifers or broadleaves has higher values, while the mix of both reaches lower $SD_t$ and is significantly different from broadleaves according to the multiple-range test.

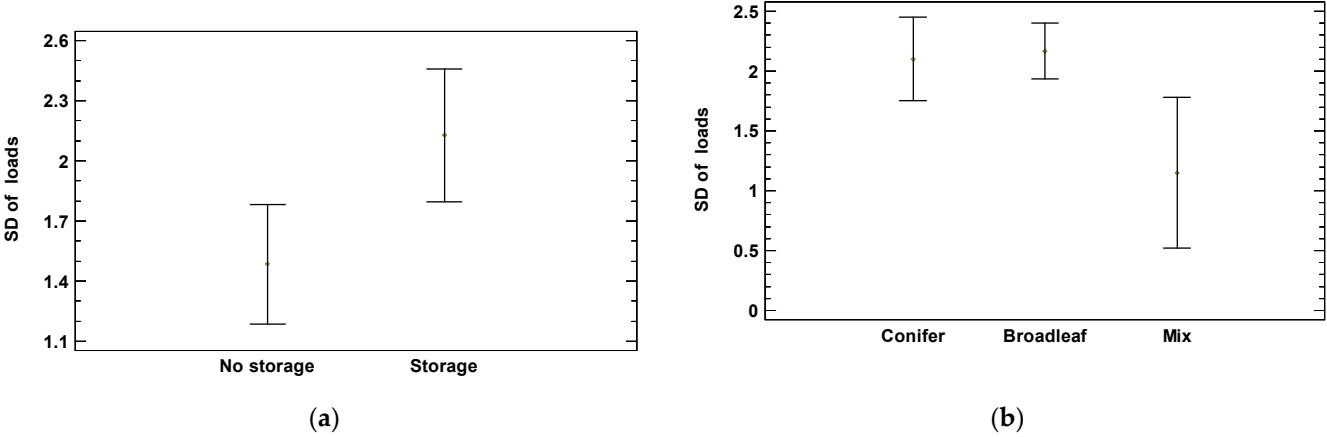

(**a**) (**b**)

**Figure 6.** Means plot of the influence of storage (**a**) and species (**b**) factors over $SD_t$ values.

### 3.2. *Estimate of Moisture Content and Sampling Intensity*

#### 3.2.1. Influence of Sampling Point

The comparison of MC values of the FAST samples collected in the five different predefined sampling points showed no significant difference (Figure 7). This result confirms that sampling may be performed on random positions of the biomass pile without compromising the reliability of the procedure. For the purpose of this study, it also endorses the use of random combinations of the five FAST values to estimate the total MC value without incurring procedural bias.

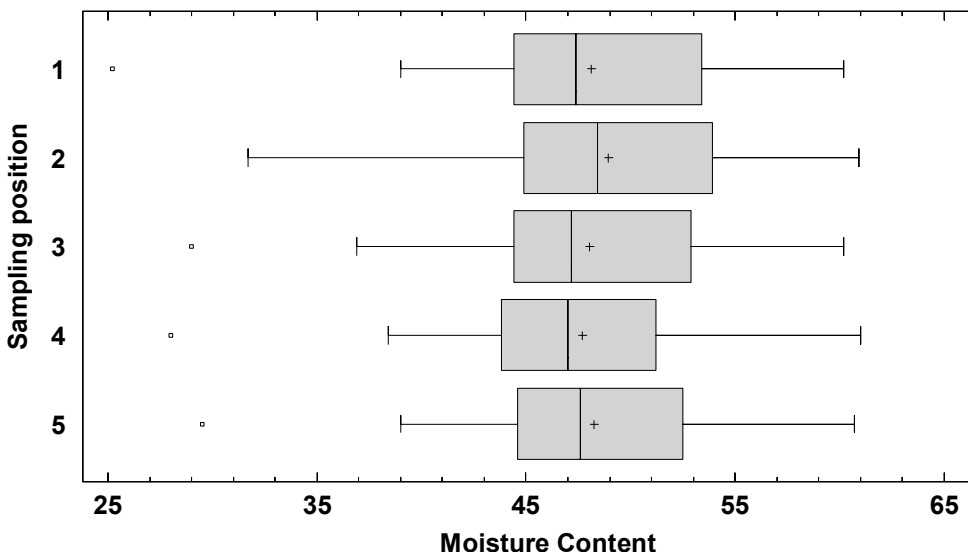

**Figure 7.** Box-plot of the five sampling points (as in figure 1). The box encloses 50% of observations, the vertical line reports the median value and "+" indicates the average value.

3.2.2. Estimate of Moisture Content in Truckloads

The sampling intensity was first run on the whole dataset comparing REF_MC with MR_MC values with no randomization of the FAST values. These were added in order of sampling. For the analysis of the truckloads was set a threshold of ±2% of tolerated error, considered as the maximum admitted level for the power plant managers. Figure 8 shows the tolerance plots and the percentage of MC estimates with the desired precision according to the sampling intensity used. Single sample estimates provide less than 70% of values with the necessary precision, and four sampling points are necessary to achieve over 90% of estimates within the tolerated error (Table 3).

The MC estimates and the relative δ values of randomized combinations of FAST values provide a more robust insight of the results of an increasing sampling intensity (Table 3). In both sampling seasons, the increment from 1 to 2 FAST samples provides an increase in precision over 10%; 3 FAST samples per load are sufficient to exceed the threshold of 90% of values within the defined limits of precision.

**Table 3.** Percentage of δ values below the threshold of ± 2 percentage points of error for the different levels of sampling intensity.

| Sampling intensity | Summer | Autumn |
|:---:|:---:|:---:|
| 1 | 74.3% | 62.9% |
| 2 | 85.7% | 88.6% |
| 3 | 91.4% | 91.4% |
| 4 | 97.1% | 94.3% |

3.2.3. Accuracy of MC Estimation in Piles

Estimation of MC on piles or large lots requires a significant number of samples. However, compared to the MC assessment on truckloads, a higher level of precision can be achieved with a relatively lower number of samples. As shown in Figures 9 and 10, the estimation of MC with an average of 20 samples achieves about 90% of values within the limit of 2% of MC difference. Lower sampling intensity leads to higher errors and is increasingly affected by the inhomogeneity of the MC of wood chips: in Summer, just a tolerance level of ±4 or above allows reducing sampling intensity to 10 FAST samples and

still gets more than 90% of observations within the threshold; in Autumn, a tolerance level of ±3 is enough to achieve the same result.

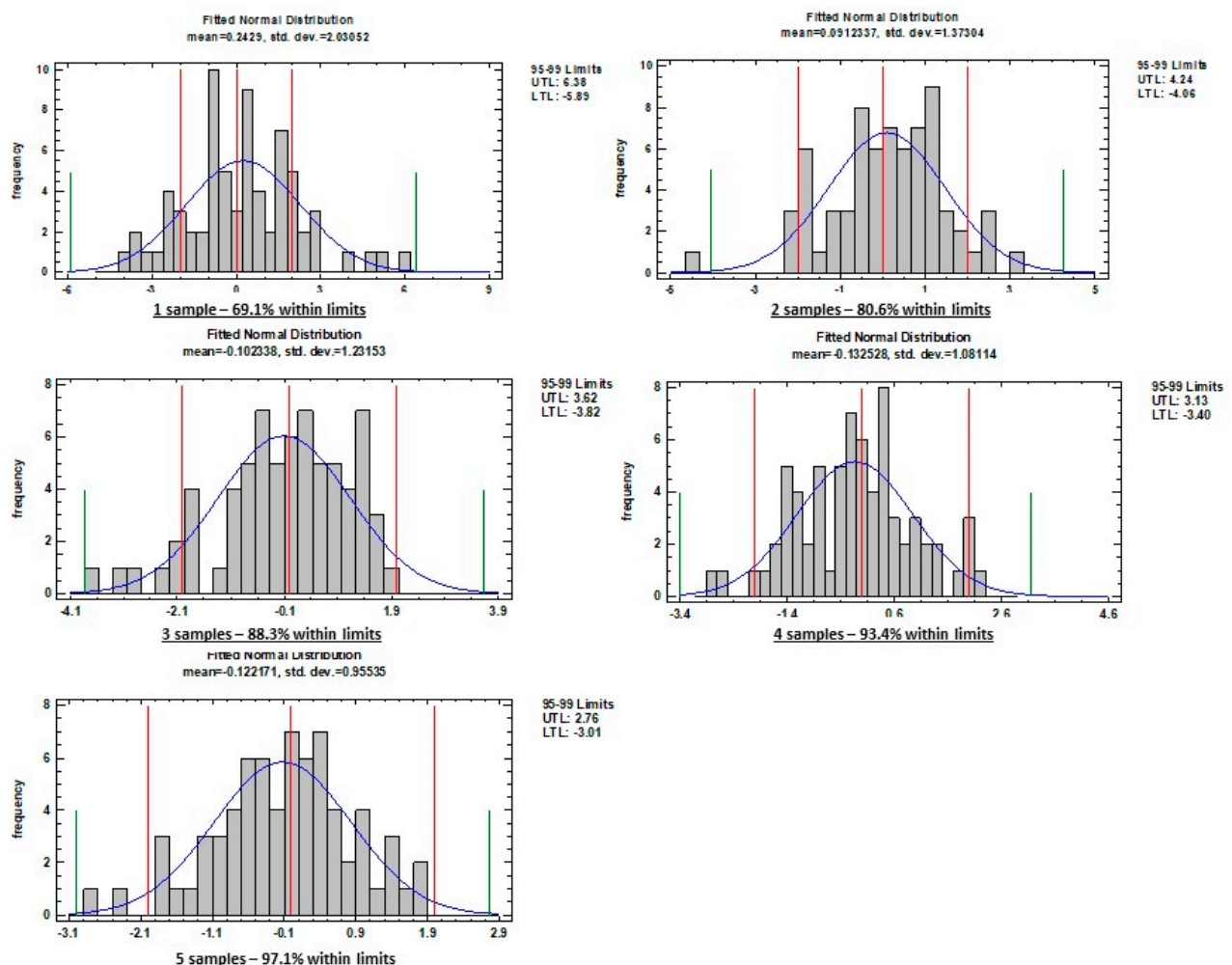

**Figure 8.** Tolerance plots of the fitted normal distribution representing the frequency of δ values (deviation of MC estimate). The central red line indicates the center of the normal distribution (exact matching). Outer red lines indicate the threshold admitted of ± 2 MC percentage points of error. The percentage of values within the limits is reported below each plot.

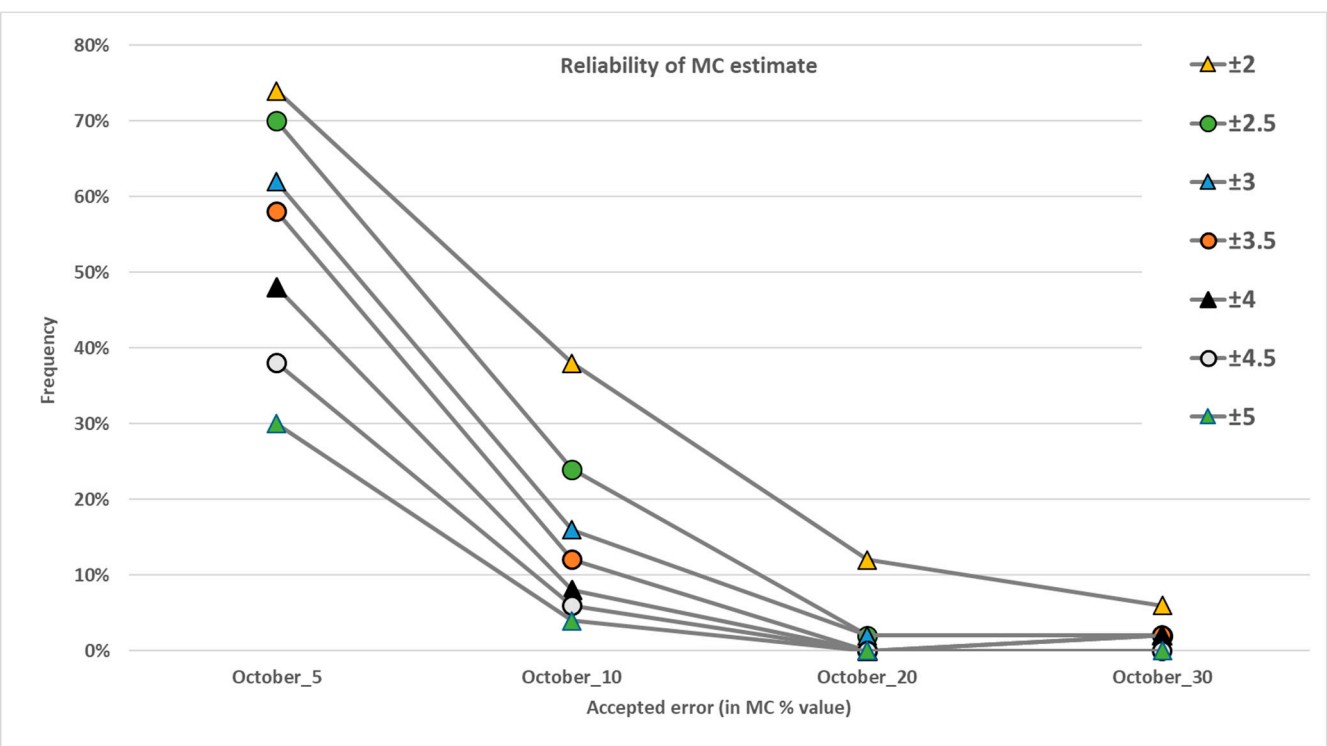

**Figure 9.** Sensitive analysis of MC estimation accuracy of the Summer pile based on the average of an increasing number of samples (5, 10, 20 and 30). Trend lines show the result according to different error tolerance levels.

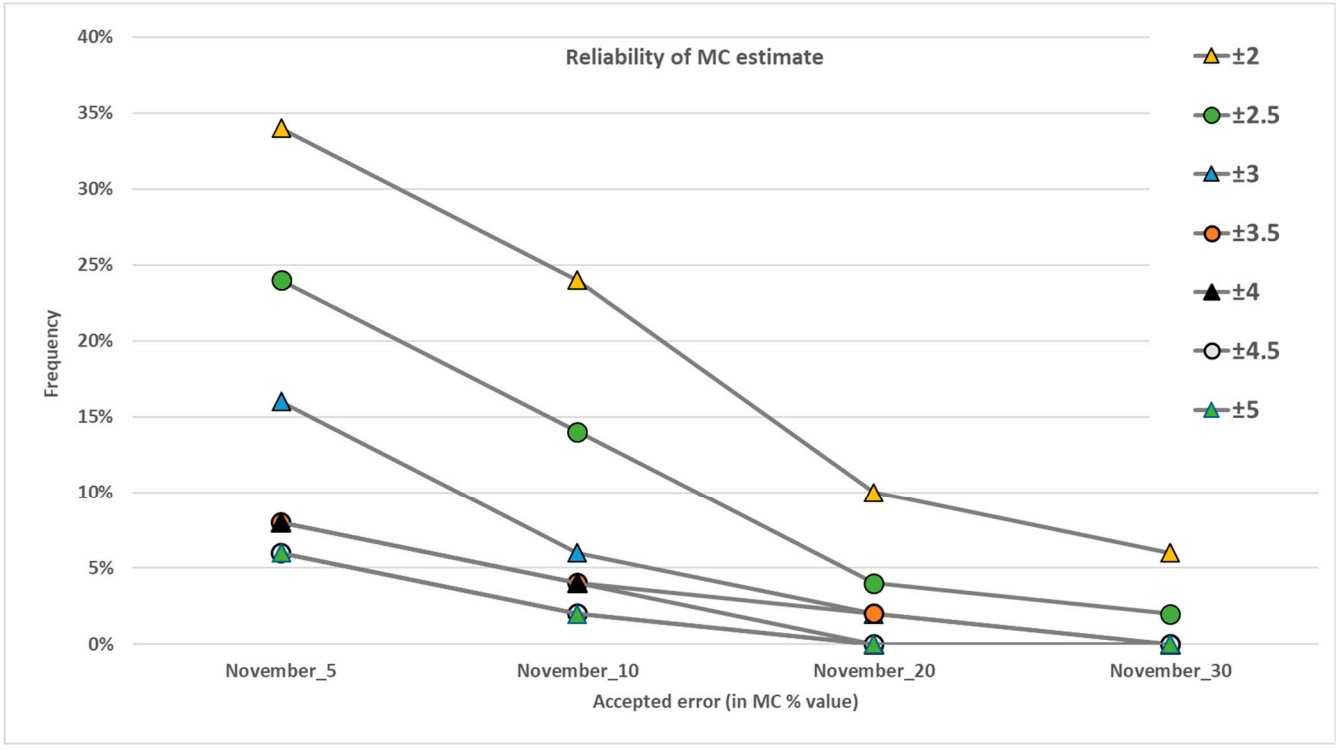

**Figure 10.** Sensitive analysis of MC estimation accuracy of the Autumn pile based on the average of an increasing number of samples (5, 10, 20 and 30). Trend lines show the result according to different error tolerance levels.

## 4. Discussion

The availability of new instruments for the determination of MC in close-to-real-time at the facility of the biomass end-users provides a unique opportunity to plan an effective control of incoming feedstock, resulting in lower fuel and yard management costs, higher

combustion efficiency and, as a consequence, lower emissions. The MC–MM analyzer tested in the study proved to be a reliable and fast instrument for the determination of MC in industrial applications, particularly thanks to the robust measurement system that is not influenced by the quality of biomass and its inherent heterogeneity.

However, in order to fully express the potential of this or comparable instruments for MC determination, it is important to reliably estimate the appropriate number of samples to be collected, i.e., the sampling intensity. This shall be determined as a function of the variability of the feedstock characteristics: a fast assessment and characterization of the incoming biomass is a key strategy to adjust the sampling intensity (and the related costs) with the desired precision of the MC estimate. Given the operative context, the estimate is necessarily based on factors that can be measured or retrieved by the biomass manager for each transport unit received. In fact, this study shows that single truckloads arriving on the same day may differ greatly in terms of internal variability, with about 35% of $SD_t$ exceeding the threshold value of 2. According to the results of this study, this variability does not depend on the season or the average MC. This outcome is consistent with other research [29,30] that highlights how, to a certain extent, the inherent and non-controllable variability of biomass hinders the efficient management of bioenergy facilities. The unique variable with a significant influence is related to the procurement system: storage of biomass. When present, this step of the supply chain increases the heterogeneity of the fuel, making it more difficult to perform an effective estimate of the MC of the unit of fuel delivered (a truckload, a barge, etc.). The importance of its correct evaluation is demonstrated by the significant influence on biomass density, affecting storage and transport conditions [6]. This is probably due to the fact that biomass producers use intermediate storage—generally uncovered—to create a buffer of feedstock, securing the provision to the end user in adverse weather conditions. This is confirmed by several studies on the effect of storage of wood chips [18–32], which leads to variations of the fuel properties, particularly if prolonged for several months. Additionally, wood chips from different forest operations are delivered and mixed at the intermediate storage, thus including several species and diverse tree sections. With this process, freshly comminuted material is mixed with "seasoned" biomass already exposed to open-air conditions, leading to a further source of variability.

It is interesting to consider that the heterogeneity observed within the load of single trucks is not related to the variability of the wood chips delivered as a whole (e.g., the daily procurement of a power plant). The case of the two sampling periods in the present study is well representative, presenting a high $SD_t$ variability with a relatively limited heterogeneity of piles.

A high and inhomogeneous $SD_t$ makes more challenging the estimate of MC of the incoming transport units, with negative consequences on biomass control, unless a high sampling intensity is adopted. MC estimated with a single sample per truck leads to a significant bias, as in the worst case (Autumn) it returns over 30% of values with an error exceeding ±2 percentage points. The adoption of two sampling points significantly enhances the reliability of the estimate, reducing to about 11–14% the values with excessive deviation. The latter can be regarded as a reasonable compromise between cost and precision. When a higher reliability is required, for instance for invoicing purposes and in order to avoid disputes, three samples should be collected in each truckload, with a ratio of about one sample every 10–11 t of feedstock. Such sampling intensity should be adopted also when the biomass manager detects in the load darker and more "seasoned" wood chips coming from intermediate storage. With three samples per truck (i.e., 1 sample every 11 t of biomass), the measuring time required by the MR–MM sets a maximum threshold of about 40–45 trucks measurable per shift—a limit that larger power plants may exceed often and that would require additional shifts or MC sensors, thus further increasing the cost of MC monitoring.

On the contrary, the MC values recorded in the piles composed by the 35 loads are relatively more homogeneous, allowing a reliable estimate of the overall MC even with a



lower number of samples. A sampling intensity of 20 samples is sufficient to achieve a probability of 90% to lay below ±2 percentage points of error in MC estimation. This corresponds to a ratio of about one sample every 57 t of feedstock. If ±3 percentage points of error are tolerated, the intensity can be reduced to 10 samples, with a significant cost reduction for the analysis and a ratio of one sample per 114 t of biomass.

## 5. Conclusions

Large bioenergy facilities are increasingly adopting fast MC measurement systems in substitution to the slow-responsive thermo-gravimetric method. Among those, Magnetic Resonance systems proved to be particularly reliable, coping with any degree of variability in terms of MC and biofuel properties. As a further benefit, these sensors allow increasing the quantity of samples measured in each load, enhancing the overall quality and reliability of the MC estimate. In order to maximize this opportunity, biomass managers should pay attention to visually evaluating the characteristics of the incoming trucks. In the case of high volume (90 m$^3$) loads with low fuel heterogeneity, two samples per truck (i.e., one sample every 16 t of biomass) may be enough to effectively control MC. However, when dealing with a higher variability (e.g., due to the presence of biomass previously stored in open-air conditions) the results of this study suggest increasing the intensity to three samples per truck (i.e., one sample every 11 t of biomass). When dealing with larger loads, such as barges, an intensity of one sample every 50–60 t of biomass appears to be adequate to correctly estimate the overall MC. The use of tables referring to different types of biofuel in ISO 18135 could be useful to guide biomass managers to define quality parameters thresholds. Yet those should be integrated with further details in order to better support biomass monitoring and quality estimates.

The results of this study relay on a relatively robust database, however further studies, possibly based on yearly records of power plants, should be conducted to better define the thresholds of variability of MC in biomass and the predictable factors influencing it, (e.g., intermediate storage practices). Such information should lay the basis for a new or integrated ISO standard sampling procedure for rapid MC gauges installed within the premises of end users, facilitating a reliable and cost effective MC estimation in commercial operations.

**Author Contributions:** Conceptualization, G.P. and E.L.; methodology, G.P., E.L. and G.A.; formal analysis, G.P., M.M., E.L.; investigation, G.A. and G.P.; data curation, G.P. and E.L.; writing—original draft preparation, E.L. and G.P.; writing—review and editing, G.P., E.L, G.A.; supervision, G.P.. All authors have read and agreed to the published version of the manuscript.

**Funding:** This research received no external funding.

**Institutional Review Board Statement:** Not applicable.

**Informed Consent Statement:** Not applicable.

**Acknowledgments:** The authors wish to acknowledge Alessandro Cinotti for his support in data collection.

**Conflicts of Interest:** The authors declare no conflict of interest.

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
