# Peer review of "Wood Fuel Procurement to Bioenergy Facilities: Analysis of Moisture Content Variability and Optimal Sampling Strategy"

_processes, doi:10.3390/pr9020359_

Round 1
Reviewer 1 Report
Thank you for submitting your work to Processes. I think that this is a sound study that can be published after a few improvements.
Here some specific suggestions:
-Form&style: check that you used the template correctly. Make sure the Figures are in a way that you want them. I don't think the editors will change them before final publication
-70 truckloads correspond to how many t?
-determine required sample per t not truckload only
-heating value in introduction. Can you provide a range for the heating value affected by the different MCs?
-ISO 18135, maybe include a SWOT analysis at the end of the introduction to provide a strong visualization of your point
-megagrams = t
-include a conclusions chapter.
-in the discussion, discuss how your new method could be used in the future industrially. I think your work is very applied, which is great. Show how this could be used on industrial scale in Italy and elsewhere and may even result in a new ISO.
Author Response
Thank you for submitting your work to Processes. I think that this is a sound study that can be published after a few improvements.
Here some specific suggestions:
- -Form&style: check that you used the template correctly. Make sure the Figures are in a way that you want them. I don't think the editors will change them before final publication
Thanks for your encouragement and the suggestions provided. Form and Style had been revised as advised as well as the format of tables.
-70 truckloads correspond to how many t?
About 2,270 tonnes of woodchips where delivered with the 70 trucks. We added this figure in abstract and Materials and methods as suggested.
-determine required sample per t not truckload only
Excellent suggestion. In the original manuscript we just mentioned that for the truckload case. We added this detail in the discussion both for truckload (3 samples = about 1 sample for 11 t of biomass) and for piles (20 samples = about 1 sample for 57 t of biomass). We did not add this figure in the abstract due to the words limits and, according to the guidelines, in order to avoid too much essential info that may discourage readers to go through the whole paper.
- -heating value in introduction. Can you provide a range for the heating value affected by the different MCs?
We added a reference of the heating value variation as suggested.
- -ISO 18135, maybe include a SWOT analysis at the end of the introduction to provide a strong visualization of your point
SWOT analysis is usually used to improve an activity management or to enhance a resource dealing, but it’s rarely used to judge the usefulness about a Standard, as in this case. Nevertheless, the attempt to carry out an objective analysis about opportunities and risk about ISO 18135 has been added in introduction. Moreover, a significative comment about feasibility of ISO 18135 is also reported in the related. citation.
- -megagrams = t
Modified as suggested throughout the manuscript
- -include a conclusions chapter.
We added a conclusion section as suggested
- -in the discussion, discuss how your new method could be used in the future industrially. I think your work is very applied, which is great. Show how this could be used on industrial scale in Italy and elsewhere and may even result in a new ISO.
Thanks for the advice. We added an explanation of the practical usage of the outcomes of this study and suggestions for future developments towards a revision of the ISO.
Reviewer 2 Report
Comments
- Line 313 - 50%, Line: 321 - 2 %, a space or no spaces to standardize throughout the work.
- Suggestion
The authors do not enter into a discussion with the results of research, but only state the fact of its existence. This is far from satisfactory in terms of good discussion. The results should be discussed with the more findings reported in recent papers.
Author Response
Line 313 - 50%, Line: 321 - 2 %, a space or no spaces to standardize throughout the work.
The format XX% has been standardized throughout the manuscript as suggested
Suggestion
The authors do not enter into a discussion with the results of research, but only state the fact of its existence. This is far from satisfactory in terms of good discussion. The results should be discussed with the more findings reported in recent papers.
Thanks for the valuable suggestion. We integrated the discussion and added a Conclusion section. We also increased the number of recent papers and compared their findings with our results. It is worth highlighting that few researches can be found focusing specifically on the control of variability, sampling procedures and the influence of both on MC estimate.
Reviewer 3 Report
The authors raise a very important and current research topic in their work. The published article will surely attract the attention of readers. The proposed research methodology is correct and the results are well developed. The drawback of the article is that it is poorly embedded in the available literature. The authors refer to less than 20 scientific works, which reduces the credibility of a good diagnosis of the current state of knowledge, although the cited works are very up-to-date. I believe that the authors need to improve this aspect in the introduction and discussion. Especially in the discussion chapter, the results of the research should be related to the work of other researchers. I believe that the article is worth publishing, but it is necessary to raise its scientific level by significantly expanding the state of knowledge in the available scientific articles.
Detailed comments: The article is characterized by solid research and important conclusions that are determined by research. The disadvantage of the Discussion chapter is the lack of references to the results of other authors, as the topic of chip moisture analysis is quite popular.
I believe the authors should extend this. References - the authors use the current literature, but in my opinion there are too few of them to show that the authors recognize the topic well. I am sending a few sample works that the authors can refer to:
Pan, P.; McDonald, T.; Fulton, J.; Via, B.; Hung, J. Simultaneous Moisture Content and Mass Flow Measurements in Wood Chip Flows Using Coupled Dielectric and Impact Sensors. Sensors 2017, 17, 20. https://doi.org/10.3390/s17010020
Pichorim, S.F.; Gomes, N.J.; Batchelor, J.C. Two Solutions of Soil Moisture Sensing with RFID for Landslide Monitoring. Sensors 2018, 18, 452. https://doi.org/10.3390/s18020452
Therasme, O.; Eisenbies, M.H.; Volk, T.A. Overhead Protection Increases Fuel Quality and Natural Drying of Leaf-On Woody Biomass Storage Piles. Forests 2019, 10, 390. https://doi.org/10.3390/f10050390
The work is in Polish, but it has a summary and captions under the drawings in English. https://www.researchgate.net/profile/Arkadiusz_Gendek/publication/324200860_Effect_of_harvest_method_and_composition_of_wood_chips_on_their_caloric_value_and_ash_content/links/5ac48957a6fdcc1a5bd05ae6/Effect-of-harvest-method-and-composition-of-wood-chips-on-their-caloric-value-and-ash-content.pdf
Gendek A., Malatak J., Velebil J. 2018. Wpływ technologii pozyskania i składu zrębków leśnych na ich wartośćopałową i zawartość popiołu. Sylwan 162 (3): 248−257. DOI: https://doi.org/10.26202/sylwan.2017125.
RUDOLF PETRÁŠ, JULIAN MECKO, DANICA KRUPOVÁ, ANDREJ PAŽITNÝ: Aboveground biomass basic density of hardwoods tree species. 1001. doi.org/10.37763/wr.1336-4561/65.6.10011012
Of course, this is only a suggestion, the authors do not have to refer to the indicated works. This is to indicate that there are works on the subject available.
Introduction Line 38 - The authors may consider whether the items in the literature below will enrich their literature due to the confirmation of the statement about municipal waste as wood biomass.
Warguła, Ł.; Kukla, M.; Lijewski, P.; Dobrzyński, M.; Markiewicz, F. Influence of Innovative Woodchipper Speed Control Systems on Exhaust Gas Emissions and Fuel Consumption in Urban Areas. Energies 2020, 13, 3330. https://doi.org/10.3390/en13133330
Line 68 - The authors suggest a system that allows humidity measurement in order to refuse to accept it, suggesting unfavorable environmental impact when burning wet material, but whether they have any idea for a different use of this raw material, if energy has already been incurred for its processing. Doesn't the use of the raw material for which energy has already been used seem less ecological than burning it with worse operating parameters? I agree that the humidity should certainly affect the price, but I'm not entirely sure if it should be the basis for a refusal of goods.
Congratulations on the work done.
Author Response
The authors raise a very important and current research topic in their work. The published article will surely attract the attention of readers. The proposed research methodology is correct and the results are well developed. The drawback of the article is that it is poorly embedded in the available literature. The authors refer to less than 20 scientific works, which reduces the credibility of a good diagnosis of the current state of knowledge, although the cited works are very up-to-date. I believe that the authors need to improve this aspect in the introduction and discussion. Especially in the discussion chapter, the results of the research should be related to the work of other researchers. I believe that the article is worth publishing, but it is necessary to raise its scientific level by significantly expanding the state of knowledge in the available scientific articles.
Thanks for the encouraging words and for the suggestions. We enhanced the discussion and comparison with other articles. It is worth highlighting that few researches can be found focusing specifically on the control of variability, sampling procedures and the influence of both on MC estimate.
Detailed comments: The article is characterized by solid research and important conclusions that are determined by research. The disadvantage of the Discussion chapter is the lack of references to the results of other authors, as the topic of chip moisture analysis is quite popular.
- I believe the authors should extend this. References - the authors use the current literature, but in my opinion there are too few of them to show that the authors recognize the topic well. I am sending a few sample works that the authors can refer to: VEDI SE QUALCUNO PUò ESSERE UTILE ANCHE PER INTEGRARE LA BIBLIOGRAFIA
- Pan, P.; McDonald, T.; Fulton, J.; Via, B.; Hung, J. Simultaneous Moisture Content and Mass Flow Measurements in Wood Chip Flows Using Coupled Dielectric and Impact Sensors. Sensors 2017, 17, 20. https://doi.org/10.3390/s17010020 -- (135 in biblio)
- Pichorim, S.F.; Gomes, N.J.; Batchelor, J.C. Two Solutions of Soil Moisture Sensing with RFID for Landslide Monitoring. Sensors 2018, 18, 452. https://doi.org/10.3390/s18020452
- Therasme, O.; Eisenbies, M.H.; Volk, T.A. Overhead Protection Increases Fuel Quality and Natural Drying of Leaf-On Woody Biomass Storage Piles. Forests 2019, 10, 390. https://doi.org/10.3390/f10050390 -- (136 in biblio)
- The work is in Polish, but it has a summary and captions under the drawings in English. https://www.researchgate.net/profile/Arkadiusz_Gendek/publication/324200860_Effect_of_harvest_method_and_composition_of_wood_chips_on_their_caloric_value_and_ash_content/links/5ac48957a6fdcc1a5bd05ae6/Effect-of-harvest-method-and-composition-of-wood-chips-on-their-caloric-value-and-ash-content.pdf Gendek A., Malatak J., Velebil J. 2018. Wpływ technologii pozyskania i składu zrębków leśnych na ich wartośćopałową i zawartość popiołu. Sylwan 162 (3): 248−257. DOI: https://doi.org/10.26202/sylwan.2017125.
- RUDOLF PETRÁŠ, JULIAN MECKO, DANICA KRUPOVÁ, ANDREJ PAŽITNÝ: Aboveground biomass basic density of hardwoods tree species. doi.org/10.37763/wr.1336-4561/65.6.10011012
Of course, this is only a suggestion, the authors do not have to refer to the indicated works. This is to indicate that there are works on the subject available.
Thanks a lot for the suggestion. We went through the suggested manuscripts and added the relevant ones in introduction. We also performed a further bibliography analysis with the aim to enrich the references, and the relevant ones were included either in introduction or discussion.
- Introduction Line 38 - The authors may consider whether the items in the literature below will enrich their literature due to the confirmation of the statement about municipal waste as wood biomass.
- Warguła, Ł.; Kukla, M.; Lijewski, P.; Dobrzyński, M.; Markiewicz, F. Influence of Innovative Woodchipper Speed Control Systems on Exhaust Gas Emissions and Fuel Consumption in Urban Areas. Energies 2020, 13, 3330. https://doi.org/10.3390/en13133330
Thanks you for the suggestion. We studied the paper, but it seems to us too specific on the performance and emission of the comminuting device rather than on the properties of biomass. Since just the latter is object of our research, we could not find an adequate use of this research in this manuscript.
Line 68 - The authors suggest a system that allows humidity measurement in order to refuse to accept it, suggesting unfavorable environmental impact when burning wet material, but whether they have any idea for a different use of this raw material, if energy has already been incurred for its processing. Doesn't the use of the raw material for which energy has already been used seem less ecological than burning it with worse operating parameters? I agree that the humidity should certainly affect the price, but I'm not entirely sure if it should be the basis for a refusal of goods.
This is an excellent statement. Most power plants have a specific optimum MC point, meaning that the combustor is designed to burn biomass with a specific MC. Variations from this value lead to exponential decrease of the performance of combustion, leading to high emissions and low energy efficiency. This is already a good reason for biomass managers to be strict about the MC of the fuel purchased. But in everyday life there is another issue: two or three decades ago most bioenergy facilities would purchase biomass per tonnes. It is easy to measure with a certified bridge scale, making invoicing quite easy. Nevertheless, this encourages contractors to procure wet biomass (the same tree is heavier if wet), since they would get more money from the same work. In some cases, they would even pour water on the wood chips load in order to increase the overall mass. Without a control, and the risk of having the load refused, this usually led in the past to a systematic increase of MC biomass well over 50%. For this reason, at present most facilities set contracts with their providers with binding MC limits and with the option to refuse the single load if its MC exceeds the threshold.
Congratulations on the work done.
Thank you very much for your comments, suggestions and advice! This is the main reward that every researcher seeks for his work, and we are no different in appreciating this prize!!
Reviewer 4 Report
As for the characteristics of biomass, the literature review was limited, the authors should take into account similar studies that have been carried out in the discussion. The authors described the subject of biomass in a limited way. I recommend that you refer to forest biomass, which also creates a great source of renewable energy, I recommend reading the article:
Nurek, T.; Gendek, A.; Roman, K.; Dąbrowska, M. The Impact of Fractional Composition on the Mechanical Properties of Agglomerated Logging Residues. Sustainability 2020, 12, 6120. https://doi.org/10.3390/su12156120
Nurek, T., Gendek, A., & Roman, K. (2018). Forest Residues as a Renewable Source of Energy: Elemental Composition and Physical Properties. BioResources, 14 (1), 6-20. https://ojs.cnr.ncsu.edu/index.php/BioRes/article/view/BioRes_14_1_6_Nurek_Forest_Residues_Renewable_Energy/6480
Tomasz Nurek, Arkadiusz Gendek, Kamil Roman, Magdalena Dąbrowska, The effect of temperature and moisture on the chosen parameters of briquettes made of shredded logging residues, Biomass and Bioenergy, Volume 130, 2019, 105368, ISSN 0961-9534, https://doi.org/10.1016/j.biombioe.2019.105368.
For example in above article the calorific value and other properties of forest residues (forest biomass) were shown. You can find information on fractional breakdown or CHONS analysis as well as correlation using ANOVA with the Duncan test. It will certainly enrich the manuscript.
Author Response
Comments and Suggestions for Authors
- As for the characteristics of biomass, the literature review was limited, the authors should take into account similar studies that have been carried out in the discussion. The authors described the subject of biomass in a limited way. I recommend that you refer to forest biomass, which also creates a great source of renewable energy, I recommend reading the article:
- Nurek, T.; Gendek, A.; Roman, K.; Dąbrowska, M. The Impact of Fractional Composition on the Mechanical Properties of Agglomerated Logging Residues. Sustainability 2020, 12, 6120. https://doi.org/10.3390/su12156120
- Nurek, T., Gendek, A., & Roman, K. (2018). Forest Residues as a Renewable Source of Energy: Elemental Composition and Physical Properties. BioResources, 14 (1), 6-20. https://ojs.cnr.ncsu.edu/index.php/BioRes/article/view/BioRes_14_1_6_Nurek_Forest_Residues_Renewable_Energy/6480
- Tomasz Nurek, Arkadiusz Gendek, Kamil Roman, Magdalena Dąbrowska, The effect of temperature and moisture on the chosen parameters of briquettes made of shredded logging residues, Biomass and Bioenergy, Volume 130, 2019, 105368, ISSN 0961-9534, https://doi.org/10.1016/j.biombioe.2019.105368.
We increased the bibliography analysis, including also your articles, thanks.
For example in above article the calorific value and other properties of forest residues (forest biomass) were shown. You can find information on fractional breakdown or CHONS analysis as well as correlation using ANOVA with the Duncan test. It will certainly enrich the manuscript.
The study was focused on the practical management of biomass plants. In this case, detailed chemical properties of biomass is not relevant as it would be impossible to effectively assess them over thousands of tonnes of wood chips received daily. Yet the suggestion may result useful for our future publications, thanks.
Round 2
Reviewer 3 Report
The article has been significantly improved and can be accepted for publication.
Reviewer 4 Report
All comments was properly improved.